# Association between Uric Acid Levels and the Consumption of Sugar-Sweetened Carbonated Beverages in the Korean Population: The 2016 Korea National Health and Nutrition Examination Survey

**DOI:** 10.3390/nu16132167

**Published:** 2024-07-08

**Authors:** Su Min Lee, Sang Yeoup Lee, Eun Ju Park, Young In Lee, Jung In Choi, Sae Rom Lee, Ryuk Jun Kwon, Soo Min Son, Jeong Gyu Lee, Yu Hyeon Yi, Young Jin Tak, Seung Hun Lee, Gyu Lee Kim, Young Jin Ra, Young Hye Cho

**Affiliations:** 1Department of Family Medicine and Biomedical Research Institute, Pusan National University Yangsan Hospital, Yangsan 50612, Republic of Korea; yisumin11@naver.com (S.M.L.); saylee@pusan.ac.kr (S.Y.L.); everblue124@hanmail.net (E.J.P.); ylee23@gmail.com (Y.I.L.); s1jungin@hanmail.net (J.I.C.); sweetpea85@hanmai.net (S.R.L.); brain6@hanmail.net (R.J.K.); soo890624@naver.com (S.M.S.); 2Department of Family Medicine, Pusan National University School of Medicine, Yangsan 50612, Republic of Korea; eltidine@hanmail.net (J.G.L.); eeugus@hanmail.net (Y.H.Y.); o3121998@hanmail.net (Y.J.T.); greatseunghun@daum.net (S.H.L.); 3Department of Medical Education, Pusan National University School of Medicine, Yangsan 50612, Republic of Korea; 4Department of Family Medicine, Pusan National University Hospital, Busan 49241, Republic of Korea; happygaru@hanmail.net (G.L.K.); yjra80@naver.com (Y.J.R.)

**Keywords:** uric acid, sugar-sweetened carbonated beverages, metabolic syndrome

## Abstract

Elevated uric acid levels are linked with obesity and diabetes. Existing research mainly examines the relationship between sugar-sweetened carbonated beverage (SSB) consumption and uric acid levels. This study explored the association between the quantity and frequency of SSB consumption and elevated uric acid levels in Korean adults. Data from 2881 participants aged 19–64 years (1066 men and 1815 women) in the 2016 Korea National Health and Nutrition Examination Survey were analyzed. Serum uric acid levels were categorized into quartiles, with the highest defined as high uric acid (men, ≥6.7 mg/dL; women, ≥4.8 mg/dL). SSB consumption was classified into quartiles (almost never, <1 cup (<200 mL), 1–3 cups (200–600 mL), ≥3 cups (≥600 mL)) and frequency into tertiles (almost never, ≤1/week, ≥2/week). Multivariate logistic regression assessed the association, with separate analyses for men and women. Increased daily SSB consumption and frequency were significantly associated with high uric acid levels in men but not in women. After adjusting for sociodemographic and health characteristics, consuming ≥3 cups (≥600 mL) of SSBs per day and SSBs ≥ 2/week were significantly associated with high serum uric acid levels in men, but this association was not observed in women. The study concludes that increased SSB intake is linked to elevated uric acid levels in Korean men, but not in women.

## 1. Introduction

Sugar-sweetened carbonated beverages are one of the most widely consumed beverages worldwide, and their consumption is steadily increasing in Korea. According to recent findings from the Korea National Health and Nutrition Examination Survey (KNHANES), the daily beverage intake (standardized, aged 1 year and older) of South Koreans has increased dramatically over the past decade, from 119 g/day in 2011 to 230 g/day in 2020. Sugar intake from beverages accounted for 12.1 g, or about 21% of the total sugar intake. Carbonated beverages and fruit and vegetable drinks were the leading contributors. In 1998, the average energy intake per person from sugary beverages was 32 kcal, but between 2007 and 2009, it more than doubled to 82.2 kcal [1]. Among processed foods, beverages were reported to be the largest contributor to sugar intake, accounting for 31.1% [2].

Hyperuricemia is caused by excessive production or impaired renal excretion of uric acid, the end metabolite of purines derived from human DNA and RNA [3]. Hyperuricemia is a major contributor to prolonged systemic inflammation in patients with gout [4], and the inflammatory response triggered by urate crystals in patients with asymptomatic hyperuricemia may contribute to the development of obesity, chronic kidney disease, diabetes, and hypertension [5]. Hyperuricemia is a clinically important condition, given its association with a variety of systemic diseases, including obesity, diabetes, metabolic syndrome, gout, and cardiovascular disease [6,7]. Sucrose, the chemical component of sugar, is a disaccharide composed of glucose and fructose. Unlike glucose, fructose can act as a precursor for intracellular uric acid production, potentially increasing serum uric acid levels [8]. Recent epidemiologic studies in Western populations have shown a significant association between sugar-sweetened beverages (SSBs) and elevated serum uric acid levels [9,10,11,12,13,14,15], and excessive fructose consumption has also been shown to increase the risk of hyperuricemia and gout [9,15,16]. A meta-analysis of five cross-sectional studies found a significant positive association between SSB intake and serum uric acid levels [17]. However, limitations of these studies include the lack of consideration of gender differences and an insufficient number of studies. In addition, existing studies were limited to evaluating the relationship between SSB intake and uric acid levels and failed to evaluate the association between SSB consumption frequency and uric acid levels.

Therefore, this study sought to investigate the association between SSB intake and frequency of consumption and elevated uric acid levels in adults using national data from the Korea National Health and Nutrition Examination Survey (KNHANES).

## 2. Materials and Methods

### 2.1. Study Population

The KNHANES is a nationwide cross-sectional survey conducted by the Korea Centers for Disease Control and Prevention to accurately assess the health and nutritional status of the population and inform national health policies. The survey randomly selects 192 households from 192 regions annually, surveying approximately 10,000 individuals aged 1 year or older. It comprises health interviews, nutritional surveys, and health examinations. The survey uses a stratified multistage clustered probability sampling method to obtain a nationally representative sample. All the participants provided written informed consent.

In this study, data were derived from the 2016 KNHANES, which included 8150 participants (3665 men and 4485 women). Among them, we considered 4750 individuals aged 19–64 years who participated in the food frequency survey. After excluding those with missing data on dependent (serum uric acid levels) or independent variables, non-respondents to the food frequency survey, individuals diagnosed with cancer or chronic kidney disease, and those with a total energy intake < 500 kcal or >4000 kcal, the final analysis included 2881 participants (1066 men and 1815 women).

The KNHANES is government-conducted research for the public welfare and, therefore, was exempt from review by the Institutional Review Board in 2016 under Article 2, Paragraph 1, Clause 1 of the Bioethics and Safety Act and its enforcement regulations. This study was approved by the Institutional Review Board of Yangsan Pusan National University Hospital, Yangsan, Korea (IRB No. 55-2023-061). This study adhered to the principles of the Declaration of Helsinki.

### 2.2. Assessment of SSB Consumption and Definition of High Uric Acid Level

Dietary intake frequency was assessed using a validated semiquantitative food frequency questionnaire (FFQ) administered to individuals aged 19–64 years. The FFQ comprised 112 food and beverage items, including SSBs such as cola, cider, and fruit-flavored carbonated drinks. For SSBs, the recent 1-year average consumption frequency was categorized as follows: almost never, once a month, 2–3 times a month, once a week, 2–4 times a week, 5–6 times a week, once a day, twice a day, and three times a day. The amount of carbonated beverages consumed per occasion was classified into the following categories: less than once a month, 0.5 cup (100 mL), 1 cup (200 mL), and 1.5 cups (300 mL).

The daily average consumption of SSBs in the recent year was calculated using the frequency of intake. First, the daily average frequency of consumption in the past year was determined based on the reported frequency of carbonated beverage intake. Subsequently, the daily average consumption of carbonated beverages in the recent year was calculated by multiplying this frequency by the average portion size for one-time consumption. The participants were categorized into four groups based on their daily average consumption of carbonated beverages: almost never, more than 0 cups but less than 1 cup (more than 0 mL but less than 200 mL), 1 cup or more but less than 3 cups (200 mL or more but less than 600 mL), and 3 or more cups (600 mL or more).

The frequency of SSB consumption was categorized into three groups based on the average frequency of consumption in the recent year: almost never, once or less per week, and twice or more per week.

Serum uric acid concentration was measured using the Hitachi Automatic Analyzer 7600-210 (Hitachi, Tokyo, Japan) via the uricase colorimetric method. High uric acid level was defined as the upper quartile for each sec (≥6.7 mg/dL for men and ≥4.8 mg/dL for women).

### 2.3. Measurement of Covariates

Demographic variables (sex, age, personal income, education, and marital status) and lifestyle factors (alcohol consumption, smoking status, and physical activity) were assessed using self-reported questionnaires. Personal income was categorized into four levels according to sex, age, and education. Marital status was categorized as married or unmarried. Alcohol consumption and smoking status were classified according to frequency. Physical activity included moderate (≥2.5 h/week) and high-intensity (≥1.25 h/week) or a combination. Hypertension was categorized as hypertensive, prehypertensive, or normal. Diabetes status was defined as diabetic, impaired fasting glucose level, or normal. High cholesterol levels were either observed or absent. Total energy and nutrient intake were calculated using a food frequency survey. Blood samples collected after 8 h fasting were analyzed within 24 h. Triglyceride, high-density lipoprotein cholesterol (HDL-C), and fasting blood glucose levels were assessed using the Hitachi Automatic Analyzer 7600-210. In addition, height, weight, waist circumference, and body mass index (BMI) were measured. Blood pressure was measured three times with the participants seated for at least 5 min, averaging the second and third readings. All health examinations were conducted by trained healthcare professionals following standardized protocols.

### 2.4. Statistical Analysis

The KNHANES is a nationwide survey, and all statistical analyses considered a sample extraction design utilizing sample extraction weights following the SPSS (Statistical Package for the Social Science) survey procedures. General characteristics of participants by sex and characteristics related to high uric acid levels of each sex are presented as mean ± standard deviation for continuous variables and as frequency and percentage for categorical variables. As sex differences are known to influence the risk of hyperuricemia, and significant differences were observed in general characteristics by sex in this study, separate analyses were conducted for men and women. Student’s *t*-test was used for continuous variables, and the chi-square test was used to assess differences in proportions between the two groups. Logistic regression analysis was used to analyze the association between the consumption and frequency of SSBs and high uric acid levels. Three different multivariate models were applied using “almost never consuming SSBs” as the reference group for both consumption and frequency, and odds ratios (ORs) with 95% confidence intervals (CIs) were calculated.

No adjusted variables were included in the baseline model (Model 1). Model 2 was adjusted for age, personal income, educational level, physical activity, hypertension status, diabetes status, hypercholesterolemia status, smoking history, alcohol consumption history, systolic blood pressure, diastolic blood pressure, waist circumference, BMI, and levels of fasting blood glucose, triglyceride, and HDL-C. Model 3 was further adjusted for total daily intake of energy (kcal/day), protein (kcal/day), saturated fat (kcal/day), and carbohydrates (g/day) in addition to the variables in Model 2. All statistical analyses were performed using IBM SPSS Statistics 25 (IBM Company, Armonk, NY, USA), with a two-tailed significance level of 0.05, considering *p*-values < 0.05 as statistically significant.

## 3. Results

### 3.1. General Characteristics of Participants

The general characteristics of the participants are presented in Table 1. The mean serum uric acid levels were 5.9 ± 0.0 mg/dL for men and 4.3 ± 0.0 mg/dL for women. There was a significant difference in the mean uric acid levels between men and women, with an approximate difference of 1.6 mg/dL (*p* < 0.001). The mean age did not significantly differ between men (40.6 ± 0.5 years) and women (41.5 ± 0.4 years) (*p* = 0.095). Various factors such as BMI, waist circumference, blood pressure, triglyceride levels, HDL-C levels, and fasting blood glucose levels were higher in men than in women (*p* < 0.001). Additionally, men exhibited higher rates of alcohol consumption, smoking, and physical activity (*p* < 0.001, *p* < 0.001, and *p* = 0.003, respectively). Dietary variables, including total energy intake, total protein intake, total saturated fat intake, and total carbohydrate intake were also higher in men (*p* < 0.001). Significant sex differences were observed in terms of educational level, marital status, and personal income (*p* < 0.001, *p* < 0.001, and *p* = 0.047, respectively).

### 3.2. Comparison of General Characteristics According to Elevated Uric Acid Levels (Top Quartile: Men, ≥6.7 mg/dL; Women ≥ 4.8 mg/dL)

The general characteristics of participants according to elevated uric acid levels are presented in Table 2. Among men with elevated uric acid levels (≥6.7 mg/dL), the mean uric acid concentration was 7.4 mg/dL, compared to 5.3 mg/dL in the group without elevated levels. In women, the mean uric acid concentration for those with elevated levels (≥4.8 mg/dL) was 5.4 mg/dL, while it was 3.8 mg/dL in the group without elevated levels. Among men, there were no significant differences in age, BMI, waist circumference, systolic blood pressure, diastolic blood pressure, triglyceride level, HDL-C level, education level, personal income, alcohol consumption, smoking status, physical activity, daily energy intake, daily protein intake, daily saturated fat intake, or daily carbohydrate intake between the groups with and without elevated uric acid levels. However, the fasting blood glucose levels were significantly different between the two groups (*p* = 0.022). Among women, there were no significant differences in age, BMI, waist circumference, triglyceride level, HDL-C level, fasting blood glucose level, education level, marital status, total energy intake, total protein intake, total saturated fat intake, or total carbohydrate intake between the groups with and without elevated uric acid levels. However, blood pressure, personal income, alcohol consumption, smoking status, and physical activity differed significantly between the two groups.

### 3.3. Association between SSB Consumption and Elevated Uric Acid Levels

The association between SSB consumption and elevated uric acid levels (top quartile: men, ≥6.7 mg/dL; women, ≥4.8 mg/dL) is shown in Table 3. An increase in daily SSB consumption was associated with elevated uric acid levels in men but not in women. In men, consuming three or more cups (600 mL or more) of SSBs per day was significantly associated with higher serum uric acid levels compared to rarely consuming SSBs, as indicated by Model 1 (OR = 2.013, 95% CI = 1.332–3.042), Model 2 (OR = 1.626, 95% CI = 1.013–2.612), and Model 3 (OR = 1.921, 95% CI = 1.159–3.184). However, no such association was observed among women in any of the models.

### 3.4. Association between SSB Consumption Frequency and Elevated Uric Acid Levels

The association between SSB consumption frequency and elevated uric acid levels (top quartile: men, ≥6.7 mg/dL; women, ≥4.8 mg/dL) is presented in Table 4. An increase in daily SSB consumption frequency was associated with elevated uric acid levels in men but not in women. In men, consumption of SSBs two or more times per week showed a significant association with higher serum uric acid levels compared to rarely consuming SSBs, as indicated by Model 1 (OR = 1.742, 95% CI = 1.199–2.532), Model 2 (OR = 1.374, 95% CI = 0.913–2.068), and Model 3 (OR = 1.551, 95% CI = 1.014–2.372). However, no such association was observed among women in any of the models.

## 4. Discussion

Using representative national data from Korea, this study revealed an association between the consumption and frequency of SSBs and elevated uric acid levels in men, considering various confounding factors. However, this relationship was not observed in women. After adjusting for all confounding factors, men who consumed three or more cups (600 mL or more) of SSBs daily were found to have a 1.9-fold higher risk of having high uric acid levels compared to that in those who rarely consumed SSBs. Additionally, men who consumed SSBs twice or more a week had a 1.6-fold higher risk of having high uric acid levels. Furthermore, the analysis indicated a significant linear correlation between the serum uric acid levels of men and both the quantity and frequency of SSB consumption. Conversely, in women, no such association was observed with either the quantity or frequency of SSB consumption.

The findings of this study align with those of several epidemiological studies, confirming the association between increased SSB consumption and elevated serum uric acid levels [9,10,11,12,13,14,15]. According to a study utilizing the National Health and Nutrition Examination Survey data in the United States, increased SSB consumption was linked to elevated uric acid levels in both men and women, with a more significant trend observed in men than in women [9]. This trend was not observed for diet soda. A study from Mexico, which performed statistical analysis by combining both men and women rather than analyzing them separately, reported an increased risk of hyperuricemia with higher SSB consumption, whereas consumption of diet soda showed no such association [14]. Studies from the United States and Thailand revealed a positive correlation between SSB consumption and hyperuricemia [12,15]. A cohort analysis in rural Korea demonstrated an increased risk of hyperuricemia with higher SSB consumption in both men and women, with a linear relationship between soda intake and serum uric acid levels observed in men but not in women [18]. However, studies examining the relationship between the frequency of SSB consumption and uric acid levels are limited. The results of the current study suggest that not only the quantity but also the frequency of SSB consumption influences uric acid levels.

The mechanism by which SSB consumption promotes hyperuricemia may be related to the high fructose content in SSBs. Previous research suggests that fructose increases the production of uric acid by enhancing the breakdown of AMP, a precursor of uric acid, via a mechanism similar to ethanol-induced uric acid elevation [3,19,20,21]. Fructose phosphorylation consumes ATP, leading to adenosine diphosphate (ADP) accumulation and restriction of ATP regeneration, ultimately serving as a substrate for uric acid formation [22]. Experimental studies on humans and animals have demonstrated short-term increases in uric acid levels following fructose consumption [19,23]. Furthermore, fructose may induce insulin resistance, impair glucose tolerance, and elevate circulating insulin levels [23], thereby indirectly increasing serum uric acid levels. Experimental studies in animals and short-term feeding experiments in humans have shown that high fructose intake is associated with insulin resistance, impaired glucose metabolism, and hyperinsulinemia [24,25]. This trend was observed in men but not in women, highlighting a potential sex-specific difference.

The influence of female hormones, particularly estrogen, is considered a crucial factor contributing to sex differences in uric acid levels [26]. After menopause, decreased estrogen production in women may reduce uric acid excretion, leading to increased uric acid levels and a higher risk of hyperuricemia [27]. Several studies in humans have reported a slight decrease in uric acid levels with estrogen, possibly due to increased uric acid excretion [28]. Epidemiological studies have interestingly demonstrated that dietary fructose correlates with negative plasma lipid profiles in men, but this association does not appear in women [29,30]. Animal studies have consistently shown that fructose affects plasma insulin levels and insulin resistance differently in males and females [31,32]. Fructose consumption caused harmful effects only in male rats, while female rats showed no such impact. As noted earlier, blood lipid levels and insulin status are associated with serum uric acid levels. Our study, combined with these observations, suggests that sex hormones or other gender-related factors could influence the metabolism of fructose or uric acid.

According to Nicholls et al., estrogen therapy in transgender men resulted in a decrease in plasma uric acid levels and an increase in urinary uric acid, suggesting that estrogen promotes the renal clearance of urate [26]. Furthermore, it has been observed that hormone replacement therapy can lower serum uric acid levels in postmenopausal women suffering from hyperuricemia [33]. An epidemiological study based on NHANES-III data found that menopause independently increases the risk of hyperuricemia. It also indicated that postmenopausal hormone therapy has the potential to lower serum uric acid levels [34]. Testosterone is also expected to affect uric acid because of its known association with purine nucleotide metabolism. Animal studies have shown that castration of male rats decreased the synthesis of nucleotides, guanosine monophosphate, and adenosine monophosphate, and reduced nucleotide car-metabolism, and these effects were reversed by testosterone administration [35,36]. Unfortunately, we were unable to measure sex hormones in our study and therefore could not analyze differences in these hormones.

Although the influence of hormones, particularly estrogen, is an important factor, other physiologic and metabolic differences between men and women may also contribute to sex differences in the effects of SSB consumption on hyperuricemia. Sex differences in body composition, dietary habits, and overall lifestyle may also contribute to these observed differences.

Hyperuricemia contributes to the development of various diseases, including atherosclerotic cardiovascular diseases, diabetes, gout, metabolic syndrome, kidney disease, and obesity [6,11,28,37]. In addition, hyperuricemia appears earlier than other metabolic disorders, indicating its role as a risk factor and predictor of chronic disease. Therefore, increased uric acid levels associated with higher SSB intake and frequency are not only associated with an increased risk of hyperuricemia, but also with an increased risk of obesity, metabolic syndrome, and a variety of chronic diseases [38].

Despite the valuable insights this study provides, several limitations should be acknowledged. Firstly, owing to the cross-sectional study design, causal relationships could not be established, and further longitudinal studies are needed to establish causality. Secondly, our study did not account for all potential confounders, such as specific medications that can affect uric acid levels, and we did not measure sex hormones such as estrogen levels in women, potentially overlooking the impact of hormones. Thirdly, as the study used an FFQ, there is a potential for recall bias among participants.

Additionally, the age range of participants was limited to 19–64 years, which may affect the generalizability of our findings to older adults who may have different risk factors. Furthermore, our study used data from the KNHANES, which primarily includes individuals of Korean ethnic origin, limiting the generalizability of our results to other ethnic groups. Ethnic differences in genetics, diet, and lifestyle can influence uric acid metabolism and the prevalence of hyperuricemia.

Moreover, the protective effect of estrogen might be more pronounced in long-term studies rather than in cross-sectional analyses like ours. Longitudinal studies could provide a better understanding of how estrogen and age interact with uric acid levels over time. Despite these limitations, our findings align with the existing literature that suggests a sex-specific response to SSB consumption and uric acid levels, underscoring the need for further research in this area.

## 5. Conclusions

This study demonstrated that an increase in the quantity and frequency of SSB consumption among Korean men was significantly associated with elevated serum uric acid levels. However, this relationship was not observed in women, suggesting potential sex-specific factors influenced by hormonal variations. Given that hyperuricemia can contribute to the development of various diseases, including hypertension, obesity, diabetes, metabolic syndrome, gout, and ischemic heart disease, excessive consumption of SSBs and increased frequency of SSB intake should be avoided.

## Figures and Tables

**Table 1 nutrients-16-02167-t001:** Baseline characteristics of the study participants.

Characteristics	Men (*n* = 1066)	Women (*n* = 1815)	*p*-Value ^(1)^
Age (years)	40.6 ± 0.5	41.5 ± 0.4	0.095
BMI (kg/m^2^)	24.6 ± 0.1	23.1 ± 0.1	<0.001
Waist circumference (cm)	85.8 ± 0.3	77.7 ± 0.3	<0.001
SBP (mmHg)	118.9 ± 0.5	112.2 ± 0.4	<0.001
DBP (mmHg)	79.6 ± 0.3	73.8 ± 0.3	<0.001
Triglyceride (mg/dL)	168.6 ± 5.4	111.6 ± 2.5	<0.001
HDL cholesterol (mg/dL)	47.8 ± 0.4	56.2 ± 0.4	<0.001
Fasting serum glucose (mg/dL)	99.8 ± 0.9	95.7 ± 0.7	<0.001
Uric acid (mg/dL)	5.9 ± 0.0	4.3 ± 0.0	<0.001
Education			<0.001
≤Elementary school	67 (4.8)	176 (8.5)	
Middle school	73 (5.0)	144 (7.6)	
High school	382 (38.4)	680 (39.8)	
≥College	543 (51.8)	814 (44.2)	
Marital status			<0.001
Married	785 (65.7)	1492 (76.4)	
Not married	281 (34.3)	323 (23.6)	
Personal income			0.047
Lowest	272 (27.3)	425 (23.1)	
Lower middle	245 (22.4)	465 (25.7)	
Upper middle	275 (24.8)	461 (26.2)	
Highest	273 (25.5)	463 (25.1)	
Alcohol consumption			<0.001
Non-drinker	35 (2.9)	161 (7.7)	
< 1 month	333 (32.1)	965 (52.3)	
2–4 month	321 (31.6)	436 (26.1)	
2≥ a week	377 (33.4)	249 (13.9)	
Smoking status			<0.001
Never smoker	274 (27.7)	97 (6.4)	
Past smoker	375 (33.1)	108 (6.5)	
Current smoker	416 (39.2)	1606 (87.1)	
Aerobic physical activity			0.003
Yes	541 (54.4)	829 (47.6)	
No	523 (45.6)	984 (52.4)	
Dietary intake			
Daily energy intake (kcal/day)	2279.2 ± 25.4	1751.1 ± 18.7	<0.001
Daily protein intake (g/day)	74.9 ± 1.1	60.7 ± 0.8	<0.001
Daily saturated fat intake (g/day)	14.6 ± 0.3	11.3 ± 0.2	<0.001
Daily carbohydrate intake (g/day)	337.1 ± 3.6	275.2 ± 2.8	<0.001

^(1)^ *p*-values were calculated via survey regression for continuous variables and via the complex sampling chi-square test for categorical variables. Values are presented as mean ± standard error or weighted N (percentage). BMI, body mass index; SBP, systolic blood pressure; DBP, diastolic blood pressure; HDL, high-density lipoprotein.

**Table 2 nutrients-16-02167-t002:** Comparison of baseline characteristics according to hyperuricemia.

Characteristics	Males (*n* = 1066)			Females (*n* = 1815)		
	No Hyperuricemia (*n* = 787)	Hyperuricemia (*n* = 279)	*p*-Value	No Hyperuricemia (*n* = 1293)	Hyperuricemia (*n* = 522)	*p*-Value
Age (years)	41.6 ± 0.6	37.9 ± 0.8	0.317	41.6 ± 0.4	41.1 ± 0.8	0.974
BMI (kg/m^2^)	24.2 ± 0.1	25.7 ± 0.3	0.382	22.6 ± 0.1	24.2 ± 0.2	0.618
Waist circumference (cm)	84.9 ± 0.4	88.5 ± 0.7	0.522	76.5 ± 0.4	80.4 ± 0.6	0.513
SBP (mmHg)	118.2 ± 0.5	120.9 ± 0.9	0.645	111.1 ± 0.4	114.8 ± 0.8	0.030
DBP (mmHg)	78.9 ± 0.4	81.8 ± 0.7	0.922	72.7 ± 0.3	76.4 ± 0.5	<0.001
Triglyceride (mg/dL)	157.5 ± 5.9	199.5 ± 11.5	0.768	103.4 ± 2.6	131.0 ± 5.9	0.543
HDL cholesterol (mg/dL)	48.7 ± 0.5	45.5 ± 0.8	0.945	56.8 ± 0.5	54.7 ± 0.7	0.202
Fasting serum glucose (mg/dL)	100.6 ± 1.0	97.4 ± 1.1	0.022	95.4 ± 0.9	96.3 ± 0.8	0.289
Uric acid (mg/dL)	5.3 ± 0.0	7.4 ± 0.0	<0.001	3.8 ± 0.0	5.4 ± 0.0	<0.001
Education			0.076			0.199
≤Elementary school	60 (5.9)	7 (2.0)		126 (8.6)	50 (8.0)	
Middle school	53 (4.8)	20 (5.3)		110 (8.4)	34 (5.5)	
High school	281 (38.1)	101 (39.5)		486 (40.2)	194 (38.9)	
≥College	392 (51.2)	151 (53.2)		570 (42.8)	244 (47.5)	
Marital status			0.008			0.068
Married	596 (68.4)	189 (58.3)		1083 (78.0)	409 (72.5)	
Not married	191 (31.6)	90 (41.7)		210 (22.0)	113 (27.5)	
Personal income			0.202			0.028
Lowest	198 (27.5)	74 (26.6)		291 (21.5)	134 (26.9)	
Lower middle	187 (23.2)	58 (20.0)		340 (27.6)	125 (21.1)	
Upper middle	210 (25.6)	65 (22.8)		341 (26.5)	120 (25.4)	
Highest	191 (23.7)	82 (30.6)		320 (24.4)	143 (26.6)	
Alcohol consumption			0.231			0.022
Non-drinker	31 (3.6)	4 (1.0)		118 (8.1)	43 (6.8)	
≤1 month	250 (32.0)	83 (32.2)		709 (54.3)	256 (47.5)	
2–4 month	239 (1.9)	82 (31.6)		299 (24.8)	137 (28.9)	
2≥ a week	267 (32.8)	110 (35.2)		165 (12.7)	84 (16.8)	
Smoking status			0.913			0.013
Never smoker	210 (27.9)	64 (27.0)		1172 (89.2)	434 (82.0)	
Past smoker	281 (33.3)	94 (32.6)		62 (5.3)	46 (9.4)	
Current smoker	295 (38.8)	121 (40.4)		57 (5.5)	40 (8.6)	
Aerobic physical activity			0.660			0.019
Yes	392 (54.0)	149 (55.6)		575 (45.4)	254 (52.9)	
No	393 (46.0)	130 (44.4)		716 (54.6)	268 (47.1)	
Dietary intake						
Daily energy intake (kcal/day)	2262.7 ± 31.0	2325.1 ± 53.1	0.165	1737.2 ± 22.3	1784.4 ± 35.5	0.238
Daily protein intake (g/day)	74.5 ± 1.4	75.9 ± 2.3	0.727	60.1 ± 1.0	62.2 ± 1.4	0.065
Daily saturated fat intake (g/day)	14.5 ± 1.4	15.0 ± 0.5	0.228	11.2 ± 0.3	11.7 ± 0.3	0.095
Daily carbohydrate intake (g/day)	336.3 ± 4.1	339.4 ± 6.9	0.553	275.6 ± 3.2	274.2 ± 5.6	0.565

BMI, body mass index; SBP, systolic blood pressure; DBP, diastolic blood pressure; HDL, high-density lipoprotein.

**Table 3 nutrients-16-02167-t003:** Multivariate-adjusted ORs (95% CIs) of high serum uric acid levels according to daily consumption of SSBs.

	Almost Never (Reference)	<1 Cup (<200 mL) OR (95% CI)	1–3 Cups (200–600 mL) OR (95% CI)	≥3 Cups (≥600 mL) OR (95% CI)	*p*-Value
Males (*n* = 1066)					
Model 1	1.000	1.191 (0.814–1.745)	1.504 (0.986–2.294)	2.013 (1.332–3.042)	0.008
Model 2	1.000	1.046 (0.692–1.583)	1.323 (0.850–2.060)	1.626 (1.013–2.612)	0.001
Model 3	1.000	1.065 (0.708–1.601)	1.403 (0.890–2.211)	1.921 (1.159–3.184)	0.001
Females (*n* = 1815)					
Model 1	1.000	1.110 (0.827–1.491)	1.092 (0.791–1.507)	1.427 (0.852–2.389)	0.566
Model 2	1.000	1.088 (0.768–1.542)	1.177 (0.800–1.731)	1.403 (0.736–2.675)	0.186
Model 3	1.000	1.094 (0.776–1.542)	1.146 (0.787–1.668)	1.414 (0.759–2.635)	0.219

OR, odds ratio; CI, confidence interval; SSB, sugar-sweetened carbonated beverage.

**Table 4 nutrients-16-02167-t004:** Multivariate-adjusted ORs (95% CIs) of high serum uric acid levels according to frequency of SSB consumption.

	Almost Never (Reference)	≤1/a Week OR (95% CI)	≥2/a Week OR (95% CI)	*p*-Value
Males (*n* = 1066)				
Model 1	1	1.342 (0.964–1.868)	1.742 (1.199–2.532)	0.015
Model 2	1	1.176 (0.830–1.667)	1.374 (0.913–2.068)	0.001
Model 3	1	1.194 (0.846–1.687)	1.551 (1.014–2.372)	0.001
Females (*n* = 1815)				
Model 1	1	1.077 (0.833–1.392)	1.345 (0.943–1.921)	0.263
Model 2	1	1.089 (0.795–1.493)	1.350 (0.865–2.107)	0.182
Model 3	1	1.086 (0.797–1.479)	1.344 (0.876–2.061)	0.213

OR, odds ratio; CI, confidence interval; SSB, sugar-sweetened carbonated beverage.

## Data Availability

Data and materials are available upon reasonable request. The raw KNHANES data used in this paper can be accessed via the following website: https://knhanes.kdca.go.kr/knhanes/sub03/sub03_02_05.do, accessed on 13 December 2023.

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
