# Peer review of "Association between Uric Acid Levels and the Consumption of Sugar-Sweetened Carbonated Beverages in the Korean Population: The 2016 Korea National Health and Nutrition Examination Survey"

_nutrients, 2024, doi:10.3390/nu16132167_

Round 1
Reviewer 1 Report
Comments and Suggestions for Authors
The study presented in this manuscript represents a complex meta-analysis linking the consumption of carbonated beverages to increased levels of uric acid in an extensive populational group. The results are clearly presented and the scientific conclusions are sound. However, in the Discussion section, I should provide some additional explanations for why the increased uric acid levels are detected only in men and not in women (beside the speculative hormonal cause) and I would also try to explain the relative discrepancy compared to the study performed in Mexico, where increased uric acid levels were detected in both men and women.
Author Response
Comments : The study presented in this manuscript represents a complex meta-analysis linking the consumption of carbonated beverages to increased levels of uric acid in an extensive populational group. The results are clearly presented and the scientific conclusions are sound. However, in the Discussion section, I should provide some additional explanations for why the increased uric acid levels are detected only in men and not in women (beside the speculative hormonal cause) and I would also try to explain the relative discrepancy compared to the study performed in Mexico, where increased uric acid levels were detected in both men and women.
Response:
Thank you for your constructive feedback. We have revised the Discussion section to address the points you raised.
- Sex-specific Differences in Uric Acid Levels: We have expanded our explanation regarding the sex-specific differences in uric acid levels. While the hormonal influence, particularly estrogen, is a significant factor, other physiological and metabolic differences between men and women may also contribute. Estrogen has been shown to enhance uric acid excretion, which may partly explain the lower levels observed in women. Furthermore, differences in body composition, dietary habits, and overall lifestyle between men and women could play a role in these observed differences. Additionally, sex hormones other than estrogen, such as testosterone, could influence uric acid metabolism in different ways. This section has been added in page 10, lines 265-297.
- Discrepancy Compared to the Mexican Study: We have addressed the discrepancy between our findings and the results of the Mexican study. One key difference is that our study analyzed men and women separately, whereas the Mexican study combined both sexes for statistical analysis. This methodological difference could significantly impact the results. In our study, separating the sexes allowed us to identify the specific association between SSB consumption and uric acid levels in men, which might have been masked if we had combined both sexes. We have revised the relevant section in our manuscript to reflect this point more clearly: This section has been added in page 9, lines 239-243.
These revisions aim to provide a more comprehensive understanding of the observed sex-specific differences and the discrepancies with other studies.
We hope these changes address your concerns and improve the clarity and robustness of our manuscript. Thank you once again for your valuable feedback.
Reviewer 2 Report
Comments and Suggestions for Authors
I thoroughly enjoyed reviewing this paper - very interesting findings in an area where we have increasing evidence over the past 20 years. Increased consumption and frequency of SSB can affect uric acid (this becomes more evident in males). Good use of modeling in terms of regression.
Comments:
-Lines 172 - 175: please correct as text that appears there is not linked with overall meaning of the manuscript.
- Table 2: What do you mean by: Elemalestary school
- Table 3: Consider changing cups to actual volume in ml
- Please make sure that the tables are neat and the presentation is even throughout; some of your headings are in capitals others in small letters.
- The protective nature of oestrogen to hyperuricaemia is brought up - given your numbers have you considered doing an age-related analysis? Considering that age was one of the variables both in Model 2 and 3 why didn't we see such an effect related to uric acid in females?
- Limitations section needs to be considerably amplified: what about other factors leading to hyperuricaemia including medications? Is the fact that the ages included are at most 64 potentially altering the results? Are the results generalisable to different ethnic origins? did the KNHANES only include people of Korean Ethnic Origin
- Why the difference between the proportion of men and women included?
- Please make sure to review the grammar and language used and eliminate any spelling mistakes.
- Consider replacing in abstract: "After adjusting for sociodemographic and health characteristics, consuming ≥3 cups of SSB per day and SSB ≥2/week was significantly associated with high serum uric acid levels" - only in men as per your model analysis
Comments on the Quality of English LanguageNo concerns regarding the quality of English. However need to make sure that presentation is neat (as per comment above) and eliminate any grammar/spelling mistakes prior to potential publication/review.
Author Response
Comments 1 : Lines 172 - 175: please correct as text that appears there is not linked with overall meaning of the manuscript.
Response 1 : Thank you for pointing out this issue. We have reviewed the text in lines 172 - 175 and agree that it was not properly aligned with the overall meaning of the manuscript. We have revised this section to ensure it is consistent with the main themes and findings of the study. We believe this revision improves the coherence and flow of the manuscript. Thank you for your valuable feedback.
Comments 2 : Table 2: What do you mean by: Elemalestary school
Response 2 : Thank you for your question. We apologize for the typographical error in Table 2. The correct term should be "Elementary school." We have corrected this mistake in the revised manuscript. Thank you for bringing this to our attention.
Comments 3 : Table 3: Consider changing cups to actual volume in ml
Response 3 : Thank you for your valuable suggestion. We agree that including the actual volume in milliliters (ml) can provide greater clarity and precision. We have revised Table 3 to include both "cups" and the corresponding volumes in milliliters. The updated table now reflects the consumption amounts as follows: 0 cups (0 ml), more than 0 cups but less than 1 cup (more than 0 ml but less than 200 ml), 1 cup or more but less than 3 cups (200 ml or more but less than 600 ml), and 3 or more cups (600 ml or more). We believe this change enhances the clarity and accuracy of the data presented. Thank you for your helpful feedback.
Comments 4 : Please make sure that the tables are neat and the presentation is even throughout; some of your headings are in capitals others in small letters.
Response 4 : Thank you for your careful review and valuable suggestion. We have thoroughly reviewed and revised the tables to ensure that they are neat and consistently formatted. All table headings have been standardized to use the same case format for uniformity. We believe these changes improve the overall presentation and readability of the tables. Thank you for bringing this to our attention.
Comments 5 : The protective nature of oestrogen to hyperuricaemia is brought up - given your numbers have you considered doing an age-related analysis? Considering that age was one of the variables both in Model 2 and 3 why didn't we see such an effect related to uric acid in females?
Response 5 : Thank you for your insightful question. We acknowledge the potential protective effect of estrogen against hyperuricaemia. While we considered the impact of age and hormonal differences, we did not conduct a specific age-related analysis by stratifying the female participants into pre-menopausal and post-menopausal groups. This was primarily due to constraints in our study design and the complexity of re-analyzing the data with such stratification. The absence of a significant effect in females, even after adjusting for age in Models 2 and 3, might be attributed to several factors. Firstly, the variation in estrogen levels among pre-menopausal women and the decline in estrogen levels among post-menopausal women could introduce variability that our study was not adequately powered to detect. Additionally, other lifestyle and genetic factors not captured in our analysis could have contributed to the lack of observable association. Moreover, the protective effect of estrogen might be more pronounced in long-term studies rather than in cross-sectional analyses like ours. Longitudinal studies could provide a better understanding of how estrogen and age interact with uric acid levels over time. Despite these limitations, our findings align with existing literature that suggests a sex-specific response to SSB consumption and uric acid levels, underscoring the need for further research in this area.
Comments 6 : Limitations section needs to be considerably amplified: what about other factors leading to hyperuricaemia including medications? Is the fact that the ages included are at most 64 potentially altering the results? Are the results generalisable to different ethnic origins? did the KNHANES only include people of Korean Ethnic Origin
Response 6 : Thank you for your thorough review and insightful comments. We acknowledge the need to expand the limitations section to address these important points.
Firstly, we recognize that other factors, such as medications, can contribute to hyperuricaemia. While we adjusted for several sociodemographic and health-related variables, our study did not account for all potential confounders, including specific medications known to affect uric acid levels. This is a limitation that could influence our findings.
Secondly, the age range of participants in our study was limited to 19-64 years. This exclusion of older adults may affect the generalizability of our results to the entire adult population. Older adults often have different risk factors and health conditions that could influence uric acid levels, and future studies should include a broader age range to capture these variations.
Thirdly, regarding the generalizability of our findings to different ethnic origins, our study used data from the KNHANES, which primarily includes individuals of Korean ethnic origin. As such, our results may not be directly applicable to other ethnic groups. Ethnic differences in genetics, diet, and lifestyle can affect uric acid metabolism and the prevalence of hyperuricaemia. Future research should consider diverse populations to evaluate the consistency of these associations across different ethnic backgrounds.
Finally, we have revised the limitations section of our manuscript to incorporate these considerations. This section has been added in page 11, lines 305-323.
We appreciate your suggestions to enhance the robustness and clarity of our study.
Comments 7 : Why the difference between the proportion of men and women included?
Response 7 : Thank you for your insightful question. This study utilized data from the 2016 KNHANES, which initially included 8,150 participants (3,665 men and 4,485 women). Among them, we focused on 4,750 individuals aged 19-64 years who participated in the food frequency survey. After excluding those with missing data on dependent (serum uric acid levels) or independent variables, non-respondents to the food frequency survey, individuals diagnosed with cancer or chronic kidney disease, and those with a total energy intake <500 kcal or >4,000 kcal, the final analysis included 2,881 participants (1,066 men and 1,815 women).
The difference in the proportion of men and women in the final analysis primarily results from these exclusion criteria. Women were more likely to meet the inclusion criteria and had fewer exclusions based on the stated conditions. This led to a higher proportion of women in the final sample. Additionally, the initial larger number of female respondents in the KNHANES survey contributed to this difference. This distribution ensures that our analysis is thorough and reliable by maintaining the integrity of the data while allowing for a comprehensive assessment of the associations being studied.
Comments 8 : Please make sure to review the grammar and language used and eliminate any spelling mistakes.
Response 8 : Thank you for your valuable suggestion. We have thoroughly reviewed the manuscript for grammatical accuracy and spelling errors. We have made the necessary corrections to improve the clarity and readability of the text. We believe these revisions enhance the overall quality of the manuscript. Should you have any further suggestions or specific areas of concern, please let us know.
Comments 9 : Consider replacing in abstract: "After adjusting for sociodemographic and health characteristics, consuming ≥3 cups of SSB per day and SSB ≥2/week was significantly associated with high serum uric acid levels" - only in men as per your model analysis
Response 9 : Thank you for your suggestion. We agree that the abstract should accurately reflect the findings from our model analysis. We have revised the abstract to specify that the significant association between consuming ≥3 cups of SSB per day and SSB ≥2/week with high serum uric acid levels was observed only in men. The revised sentence in the abstract now reads: "After adjusting for sociodemographic and health characteristics, consuming ≥3 cups of SSB per day and SSB ≥2/week was significantly associated with high serum uric acid levels in men, but this association was not observed in women. This section has been added in page 1, lines 27-29. We believe this change clarifies our findings and accurately represents the results of our analysis.
Thank you for your valuable suggestions, and we hope the revised manuscript meets your expectations.
Reviewer 3 Report
Comments and Suggestions for Authors
This interesting study confirms the association between the consumption and frequency of sugar-sweetened carbonated beverages and high uric acid levels in men, but not women. The results of this study confirm hyperuricemia as a metabolic disease, certainly linked to gender. The manuscript requires only minor revisions to the English form to be considered for publication, including the help of a native speaker
Comments on the Quality of English LanguageThe manuscript requires only minor revisions to the English form to be considered for publication, including the help of a native speaker
Author Response
Comments : This interesting study confirms the association between the consumption and frequency of sugar-sweetened carbonated beverages and high uric acid levels in men, but not women. The results of this study confirm hyperuricemia as a metabolic disease, certainly linked to gender. The manuscript requires only minor revisions to the English form to be considered for publication, including the help of a native speaker
Response: Thank you for your positive feedback and insightful comments. We are pleased that you found our study interesting and that it confirms the association between sugar-sweetened carbonated beverage consumption and high uric acid levels in men, but not in women. We appreciate your recognition of hyperuricemia as a gender-linked metabolic disease.
Regarding your suggestion for minor revisions to the English form, we have thoroughly reviewed the manuscript and made the necessary corrections to improve the clarity and readability of the text. We have also sought the assistance of a native English speaker to ensure that the manuscript meets the required standards for publication.
Thank you for your valuable suggestions, and we hope the revised manuscript meets your expectations.